# Lichen mimesis in mid-Mesozoic lacewings

Hui Fang[1,2], Conrad C Labandeira[1,2,3], Yiming Ma[1], Bingyu Zheng[1], Dong Ren[1], Xinli Wei[4]*, Jiaxi Liu[1]*, Yongjie Wang[1]*

[1]College of Life Sciences and Academy for Multidisciplinary Studies, Capital Normal University, Beijing, China; [2]Department of Paleobiology, National Museum of Natural History, Smithsonian Institution, Washington DC, United States; [3]Department of Entomology, University of Maryland, College Park, United States; [4]State Key Laboratory of Mycology, Institute of Microbiology, Chinese Academy of Sciences, Beijing, China

**Abstract** Animals mimicking other organisms or using camouflage to deceive predators are vital survival strategies. Modern and fossil insects can simulate diverse objects. Lichens are an ancient symbiosis between a fungus and an alga or a cyanobacterium that sometimes have a plant-like appearance and occasionally are mimicked by modern animals. Nevertheless, lichen models are almost absent in fossil record of mimicry. Here, we provide the earliest fossil evidence of a mimetic relationship between the moth lacewing mimic *Lichenipolystoechotes* gen. nov. and its co-occurring fossil lichen model *Daohugouthallus ciliiferus*. We corroborate the lichen affinity of *D. ciliiferus* and document this mimetic relationship by providing structural similarities and detailed measurements of the mimic's wing and correspondingly the model's thallus. Our discovery of lichen mimesis predates modern lichen-insect associations by 165 million years, indicating that during the mid-Mesozoic, the lichen-insect mimesis system was well established and provided lacewings with highly honed survival strategies.

*For correspondence:
weixl@im.ac.cn (XW);
liu-jiaxi@263.net (JL);
wangyjosmy@foxmail.com (YW)

**Competing interests:** The authors declare that no competing interests exist.

## Introduction

Modern insects have dramatic morphological specializations that match various objects of the environment. For instance, the specializations occurring in katydids and butterflies that mimic leaves, stick insects and inchworms that resemble twigs, and orchid mantids that duplicate orchid flowers, provide ecological insights for understanding mimetic associations between insect mimics and their plant models (*Stevens, 2011*; *Gullan and Cranston, 2014*; *Maran, 2017*). These and other fascinating cases reveal that mimesis or camouflage is highly effective when cryptic insects resemble closely the appropriate self-similar background, indicating the complexity of ecological relationships between insect mimics and their imitating models. When and how insects first evolved such an ingenious survival strategy is unclear. A Permian katydid exhibiting a mimicking pattern of wings similar to the modern relatives was considered the oldest case of insect mimicry (*Garrouste et al., 2016*). However, evidence for a contemporaneous mimetic relationship in this Permian deposit was scarce, and there was no quantitative or other useful data to track the mimetic interactions among the insect, model and predator. More recent cases of insect mimicry have been recorded from the Mesozoic, indicating the existence of several such effective survival strategies. As in morphological specializations involving masterly deceit found in modern insects, several Mesozoic insect taxa developed remarkable structural adaptations resulting in highly accurate resemblances to co-existing models (*Wang et al., 2012*; *Wang et al., 2014*; *Yang et al., 2020*). Prominent among these mimetic insects are Neuroptera (lacewings, antlions and relatives), a nonspeciose relic order consisting of ca. 6000 extant species that engaged in several, impressive instances of mimicry that reveal novel and specialized strategies of deception, of which many are absent today. Striking examples are the Jurassic lacewing *Bellinympha* (Saucrosmylidae), a compression fossil, mimicking cycadophyte leaves

**eLife digest** Many insects mimic other organisms or use camouflage to hide from predators. For example, some modern animals mimic the organism lichens, which are formed from algae and fungus, and grow almost everywhere on Earth, from the Arctic to the desert.

The most iconic example of an insect mimicking a species of lichen is the peppered moth. During the industrial revolution, darker colored moths were better at surviving. But when the revolution ended and pollution levels declined, species of lichen began to re-emerge and increase the survival of paler moths. Yet, it is unclear how and when insects first evolved this ingenious survival strategy, as distinctive examples of insects mimicking lichens are missing from fossil records.

To answer this question, Fang et al. set out to find fossils of lichen-mimicking insects that occurred at the same time as fossils of lichens. This approach led to the discovery of two new species of lacewing insects and their related species of foliose lichen. Previous work suggested that the foliose lichen, which has a lobe like shape, did not exist more than 65 million years ago. However, the findings of Fang et al. indicate that the foliose lichen existed 165 million years ago during the age of dinosaurs, and therefore arose much earlier than previously thought.

The two new species found in north-eastern China, form a new subgroup within the moth lacewing family that Fang et al. have named 'Lichenipolystoechotes'. Close examination of both species of lacewing and the lichen under the microscopy revealed a near perfect match in their appearance. The branching patterns of the insects' wing markings fit the branching patterns of the lichen. Taken together, these findings suggest that, not only did lichen mimics exist in the age of the dinosaurs, but that this strategy of using lichen mimicry as a form of survival was already very effective during this time period.

This discovery suggests that, 165 million years ago, a micro-ecosystem of lichens and insects existed in north-eastern China. It invites new questions about how that ecosystem worked. For example, how did the lichen benefit from its relationship with lacewing insects? Further investigations could reveal the answers and uncover more interesting insects hidden in the fossil record.

(*Wang et al., 2010b*), and larvae of the green lacewing *Phyllochrysa* (Chrysopidae) from amber, modified to resemble co-occurring liverworts (*Liu et al., 2018*). Besides mimicry, other deceptive modes of appearance have been documented among Mesozoic lacewings, such as camouflaged larvae of chrysopid (green lacewing) and myrmeleontoid (antlion relative) neuropterans in amber, which evolved distinctive debris-carrying behaviors to enhance their predatory effectiveness (*Pérez-de la Fuente et al., 2012*; *Pérez-de la Fuente et al., 2018*; *Wang et al., 2016*; *Badano et al., 2018*). These cases collectively have promoted understanding of the early evolution of insect mimicry, but also have revealed that the currently species poor Neuroptera had evolved a significant repertoire of specializations involving morphologies and behaviors that adapted to a variety of Mesozoic settings.

In this report, we found an exceptional system of the first lichen mimesis by a fossil lacewing. These occurrences are from the Daohugou 1 locality of Inner Mongolia in northeastern China. The new lichen-like-mimicking insects represent a new genus with two new species and exhibit remarkable wing patterns that accurately resemble the contemporaneous lichen species *Daohugouthallus ciliiferus* Wang, Krings *et* Taylor, 2010 (*Wang et al., 2010a*). The lichen affinity of the *D. ciliiferus* model previously was doubted due to the absence of evidence for fungal and algal connections that would indicate the presence of lichenization and thus the presence of a mutualistic symbiosis (*Honegger et al., 2013*; *Lücking and Nelsen, 2018*). Our SEM results corroborate the actual presence of hyphae connected to algal cells on the *D. ciliiferus* specimens, indicating the foliose and sub-fruticose lichen growth forms were in existence during the Middle Jurassic. Present-day lichen-mimicking insects are widely recorded among several diverse orders, especially Coleoptera (beetles), Lepidoptera (moths and butterflies) and Orthoptera (grasshoppers, katydids and crickets), which have evolved unusual specializations of morphology and behavior consistent with co-occurring lichens and other habitation- or appearance similar organisms such as liverworts (*Gerson, 1973*; *Lücking, 2001*; *Capinera, 2008*; *Cannon, 2010*; *Lücking et al., 2010*). An extraordinary orthopteran, the lichen dragon katydid from the modern Ecuadorian Andes, provides an excellent disguise

of lichens (*Braun, 2011*). Other predatory and extant chrysopid larvae have a body mask adorned with affixed lichen fragments, an example of aggressive mimicry or the 'wolf in sheep's clothing' syndrome (*Skorepa and Sharp, 1971*; *Slocum and Lawrey, 1976*; *Wilson and Methven, 1997*; *Tauber et al., 2014*). Importantly, lichen-mimetic or -camouflaged insects have established a specialized lichen-association for feeding or sheltering to obtain survival advantage (*Gerson, 1973*). Our finding documents the earliest lichen-mimicking insect and reveals that this strategy of mimicry among insects has been in existence for minimally 165 Mya. This ancient association also will provide new insight for exploring the predator–prey relationships among insects and lichens, and the role of habitat during mid-Mesozoic time.

## Results

### Reanalysis of the previously suspected lichen fossil *Daohugouthallus ciliiferus*

We have studied five fossil lichen specimens, PB23120, B0474, B0476P/C, CNU-LICHEN-NN2019001 and CNU-LICHEN-NN2019002P/C, all of which were collected from the Daohugou 1 locality of Inner Mongolia. The newly collected specimens were identified to be *Daohugouthallus ciliiferus* based on careful observations of its distinctive morphology.

> Genus *Daohugouthallus* Wang, Krings *et* Taylor, 2010.
> Species *Daohugouthallus ciliiferus* Wang, Krings *et* Taylor, 2010.

### Emended diagnosis

The diagnosis is as follows and adds to the previous assessment (*Wang et al., 2010a*). Thallus foliose to subfruticose; lobes ca. 20–30 mm long, irregularly branched, margin sometimes revolute; lateral and terminal branches ca. 0.5–5.0 mm long, tips tapered; upper surface smooth, partly broken; aggregated black spots often present, punctiform; cilia sometimes present near the branch tips, forming filiform appendages; lobules occasionally present (*Figure 1*).

Besides the external morphology, certain anatomic characters also were determined. Upper cortex conglutinate, comprising one cell layer, very thin, ca. 1 µm thick (*Figure 2A*); algal cells globose to near globose, one-celled, mostly 1.5–2.1 µm in diameter, some in framboidal form, interconnected (*Figure 2A,B,F*) by or adhered (*Figure 2C–H*) to the fungal hyphae with simple wall-to-wall mycobiont-photobiont interface; fungal hyphae filamentous, some shriveled, septate (*Figure 2B,C, G,H*), 1.2–1.5 µm wide. These additional features (*Figure 2*) support the above diagnoses that this specimen is a fossil lichen.

### Remarks

This adpression lichen fossil was reported by *Wang et al., 2010a* as a new genus and new species of lichen, that is *Daohugouthallus ciliiferus*. However, there were no anatomic characters including both

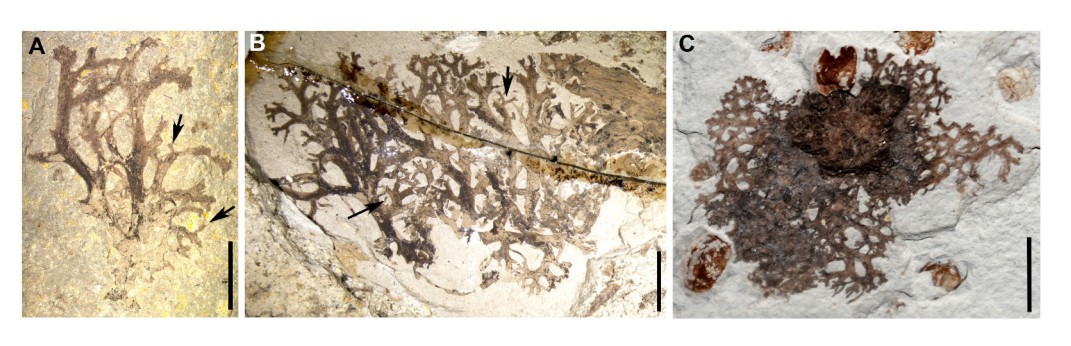

**Figure 1.** Photos of the lichen *Daohugouthallus ciliiferus* Wang, Krings *et* Taylor, 2010. (**A**) Specimen B0476P, with arrows indicating the lobules. (**B**) Specimen CNU-LICHEN-NN2019001, with arrows indicating the lobules. (**C**) Specimen CNU-LICHEN-NN2019002P. Scale bars: 5 mm in A–C.

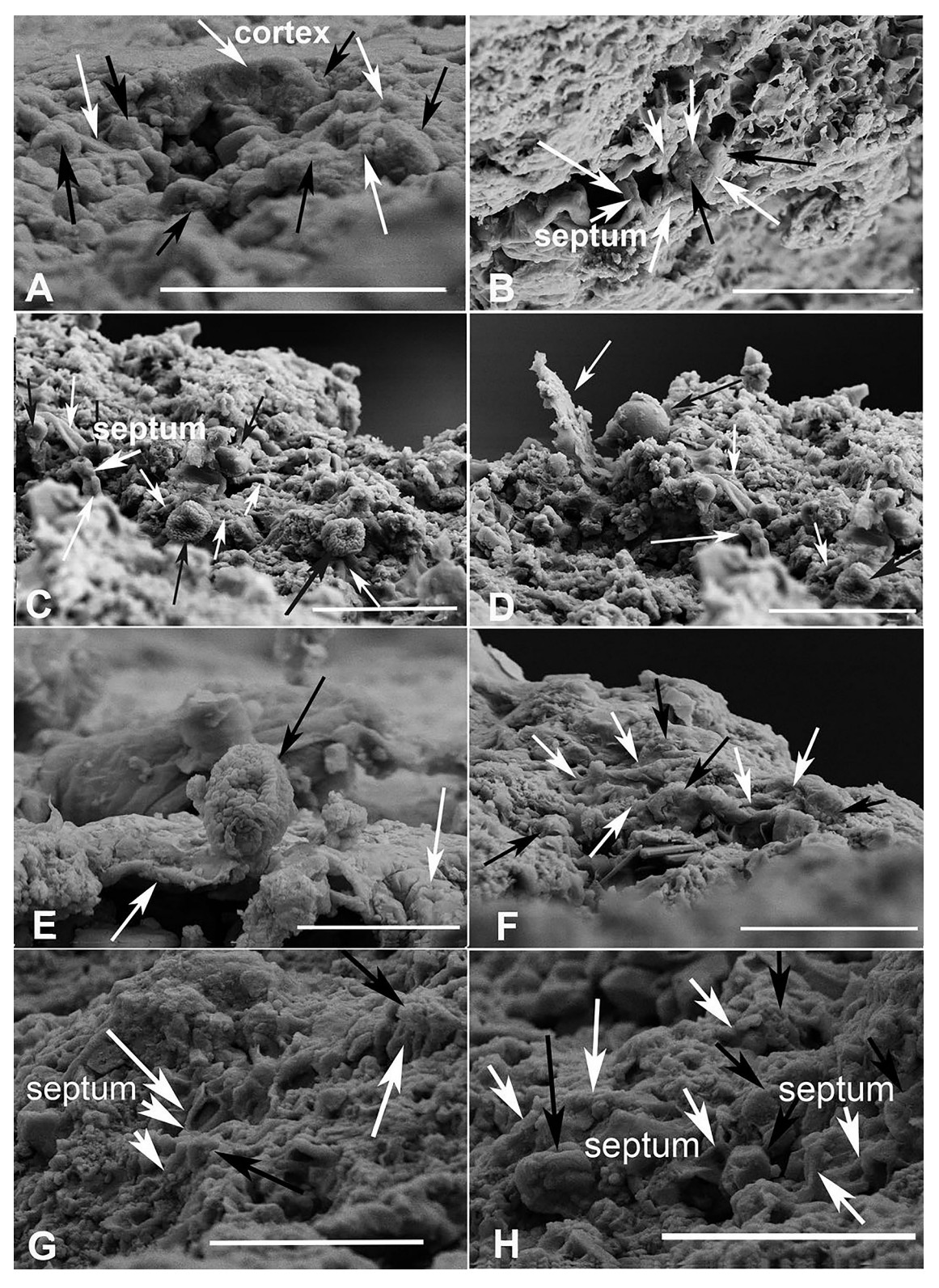

**Figure 2.** Scanning electron microscopy (SEM) micrographs of lichen fossil (CNU-LICHEN-NN2019001). (**A**) Thallus longitudinal section containing the cortex, with white arrows pointing to the fungal hyphae, and black ones to the algal cells. The fungal hyphae are interweaved with algal cells. (**B–D, F–H**) Fungal hyphae indicated by white arrows; algal cells are indicated by black arrows showing entanglement and encirclement by fungal hyphae; septa

*Figure 2 continued on next page*

*Figure 2 continued*

shown in B, C, G, H. (E) One algal cell indicated by the black arrow, displaying adherence to other fungal hyphae indicated by the white arrow. Scale bars: 5 μm in A, C, D, G, H; 10 μm in B; 3 μm in E; 4 μm in F.

fungal and algal components that was provided and consequently its lichen affinity was doubted and thought as ambiguous (*Honegger et al., 2013*; *Lücking and Nelsen, 2018*). Actually, the lichen fossil now has been well defined and should accord with three important criteria: presence of a mycobiont component, presence of a photobiont component, and presence of spatial connections between both components (*Lücking and Nelsen, 2018*). Accordingly, thallus sections were made in this study and relevant anatomic details can be observed. First, the upper cortex occasionally was present (*Figure 2A*), and the septa of fungal hyphae also is documented (*Figure 2C,D*). Second, the algal cells are globose and occasionally have a spherical assembly of microcrystals in framboidal form similar to *Trebouxia* of *Chlorolichenomycites salopensis* in morphology but much smaller (*Honegger et al., 2013*). Third, the spatial connections between fungal hyphae and algal cells have been observed, mostly consisting of fungal hyphae interweaved with algal cells (*Figure 2A,C,D,F*). The above-mentioned characters of *Daohugouthallus ciliiferus* accords well with the definition of lichen fossil and indicate a strong affinity to a lichen. From the external morphology, *Daohugouthallus ciliiferus* would be easily associated with extant *Everniastrum cirrhatum*, a conclusion that requires further study in the near future.

## Systematic paleontology

The lichen-mimicking insects represent a new genus and two new species affiliated to Ithonidae of the order Neuroptera. The terminology of venation follows *Breitkreuz et al., 2017*.

> Order Neuroptera Linnaeus, 1758
> Family Ithonidae Newman, 1853 *sensu* Winterton *et* Makarkin, 2010
> Genus *Lichenipolystoechotes* Fang, Zheng *et* Wang, gen. nov.

### Included species

*Lichenipolystoechotes angustimaculatus* Fang, Zheng *et* Wang, sp. nov. (type species), *Lichenipolystoechotes ramimaculatus* Fang, Ma *et* Wang, sp. nov.

### Etymology

The new genus name is a combination of *lichen* and *Polystoechotes* (a genus name of Ithonidae) in reference to the lichen-mimesis of the genus. The gender is masculine.

### Diagnosis

Forewing ellipsoidal shaped, medium length, slightly narrow with length-width ratio 3.25–3.5; membrane bearing coralliform pattern with unclosed diaphanous U-shaped fenestrae along the margin in forewing; costal space slightly broad basally, then narrowed towards wing apex; ScA and recurved humeral veinlet present; costal cross-veins simple in proximal half, and then distally forked; ScP and RA fused distally, ending close to the wing apex, no cross-veins present in this area; cross-veins in area between RA and RP scattered; RP with about 18 branches, RP1 a single branch, few cross-veins scattered at the radial sector; M forked beyond the separation point of RP1, MA and MP with the similar branched pattern, the number of MP branches slightly more than MA; CuA distinctly multiforked, with 7–10 pectinate branches; CuP bifurcated.

### Remarks

The new genus can easily be assigned to Ithonidae by the following characters: medium body size, prolonged forewing, relatively narrow costal space, and presence of ScA and recurrent humeral veinlet. In addition, its forewing characters, including Sc and R1 fused distally, few cross-veins except for a row of well-defined outer gradate series in radial sector, MP forked beyond MA divergence, conforming to a polystoechotid affiliation (*Zheng et al., 2016*). It also is distinguished from other polystoechotid genera by the distinctive coralliform markings of the forewings.

## *Lichenipolystoechotes angustimaculatus* Fang, Zheng *et* Wang, sp. nov. (*Figures 3E–H* and *4A,G*)

### Etymology
The specific name comes from the Latin words '*angusta*' and '*macula*' referring, respectively, to the narrow, linear and pigmented swaths on the forewing, and the spot-like patterns present on those swaths.

### Material
#### Holotype
CNU-NEU-NN2016040P/C (*Figures 3A–C* and *4E*), *paratype*. CNU-NEU-NN2016041 (*Figure 3D*).

### Type locality and horizon
Daohugou 1, near Daohugou Village, Shantou Township, Ningcheng County, Inner Mongolia, China. Jiulongshan Formation, Callovian–Oxfordian boundary interval, latest Middle Jurassic.

### Diagnosis
Forewing, humeral veinlet strongly recurved; ScA present; cross-veins in area between RA and RP scattered except for the middle gradate series; RP with about 18 branches; MA and MP with similarly distal pectinate branches; CuA pectinately branched in distal half; CuP deeply bifurcated at anterior half.

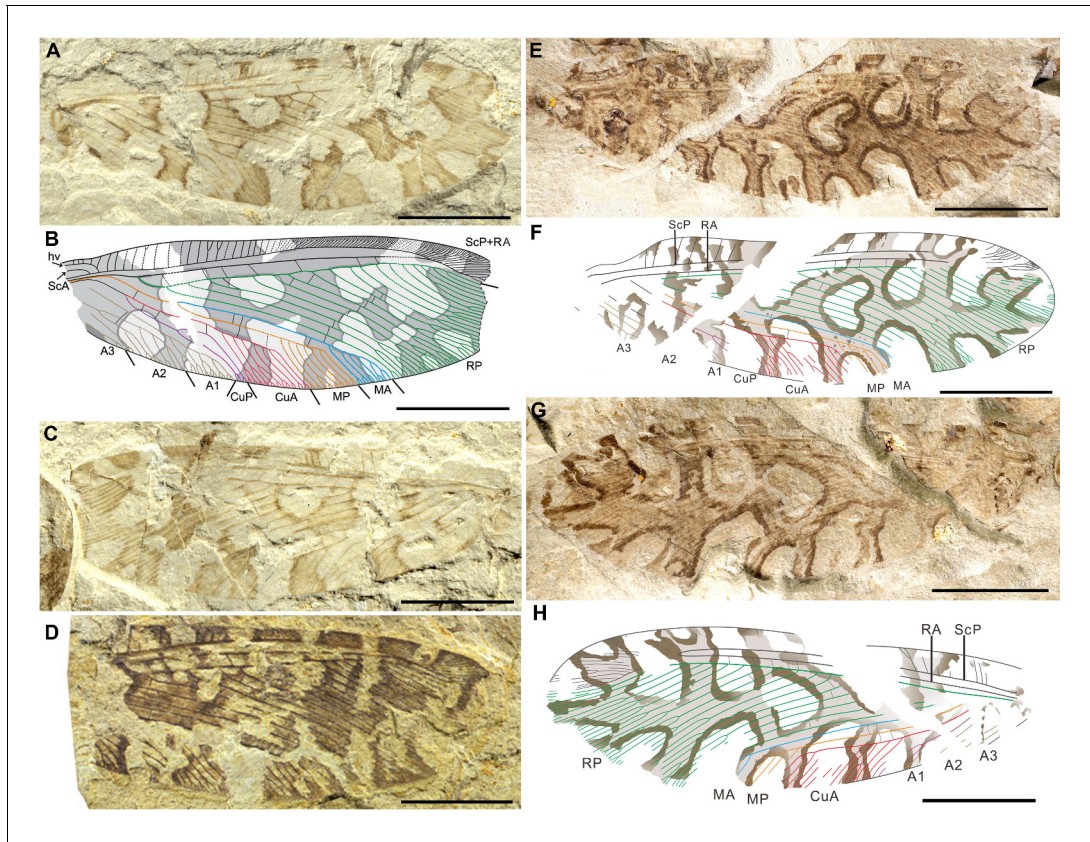

**Figure 3.** Photos and line drawings of *Lichenipolystoechotes angustimaculatus* gen. *et* sp. nov., and *L. ramimaculatus* gen. *et* sp. nov. (A–C) Holotype CNU-NEU-NN2016040P/C of *L. angustimaculatus*, photo of part in (A). Accompanying overlay drawing in (B). Photo of counterpart in (C). (D) Photo of the paratype CNU-NEU-NN2016041 of *L. angustimaculatus*. (E–H) The holotype CNU-NEU-NN2019006P/C of *L. ramimaculatus*, with a lichen mimicking forewing pattern. Photo of part in (E); accompanying overlay drawing in (F); photo of counterpart in (G); and accompanying overlay drawing in (H). Scale bars: 5 mm in A–H.

## Description and comparison

Only forewing present. Forewing elongate, oval shaped, about 21.3 mm long, 6.5 mm wide; membrane bearing irregular coralloid markings of pigmentation, forming many diaphanous marginal fenestrae; costal space slightly broad basally, then narrowed towards the wing apex; costal cross-veins scarcely branched in proximal half of wing, and then forming bifurcated branches in distal half; sc-ra cross-vein absent; space between RA and RP relatively narrow with seven cross-veins; RP with 18 pectinate branches, and each branch bifurcated near wing margin; cross-veins in radial sector relatively scarce except for the middle gradate series; M forked slightly beyond the separation of RP1; MA and MP forming seven pectinate branches each; CuA pectinate branched in distal half part, forming seven pectinate branches; CuP first bifurcated at the proximal half, then forming the distal simple forks; A1–A3 forming several pectinate branches; a few cross-veins present among MA and A3.

### *Lichenipolystoechotes ramimaculatus* Fang, Ma *et* Wang, sp. nov. (*Figures 3E–H* and *4A,G*)

#### Etymology

The specific name comes from the Latin word *rami* and *macula*, referring, respectively, to the narrow, branched and pigmented swaths traversing the forewing, and the spot-like patterns present on those branched swaths.

#### Material

##### Holotype

CNU-NEU-NN2019006P/C (*Figures 3E–H* and *4A,G*).

#### Type locality and horizon

Daohugou 1, near Daohugou Village, Shantou Township, Ningcheng County, Inner Mongolia, China. Jiulongshan Formation, Callovian–Oxfordian boundary interval, latest Middle Jurassic.

#### Diagnosis

The marginal diaphanous fenestrae significantly open, surrounded by pigmented zones; MA forming the distal dichotomizing fork, MP with six pectinate distal branches; CuA branched nearly at the middle, forming about 11 pectinate branches, CuP bifurcated beyond the middle portion of the vein.

#### Description and comparison

Only forewing preserved. Forewing elongate, oval shaped, about 22.8 mm long, 6.5 mm wide; costal space slightly broadened, then narrowed towards wing apex; subcostal veinlets relatively widely spaced, scarcely branched medially, forming multiple bifurcated branches distally; areas between Sc and RA narrow, without crossveins; space between RA and RP relatively narrow with sparse cross-veins; RP with about 22 pectinate branches; RP1 branched from RP near wing base, single until wing margin; M forked basally, MA forming two distal dichotomous branches, and MP forming six distal pectinate branches; CuA pectinate medially to distally, forming 11 pectinate branches; CuP bifurcated at middle; A1–A3 partly preserved.

## Discussion

The two new species of *Lichenipolystoechotes* exhibit a very similar appearance, but they easily can be separated by the distinct differences of branches of the MA and CuA veins. *Lichenipolystoechotes* species are conspicuous based on their highly prominent, homologous, pigmentation pattern of their forewings, which implies that these insects evolved a similar defensive strategy. The closest extant relatives of *Lichenipolystoechotes* are Ithonidae (moth lacewings), of which their ecological and biological features are poorly documented (*New, 1989*). The forewings of the two new species demonstrate a high similarity in their overall appearance, such as the forewing branching pattern (*Figures 3A,E* and *4A*) that matches the thallus branches of the co-occurring foliose to subfruticose lichen *Daohugouthallus ciliiferus* (*Figure 4B–D*; *Wang et al., 2010a*). The entire forewing forms an irregular branching pattern amid rounded, diaphanous fenestrae (windows) that are distributed

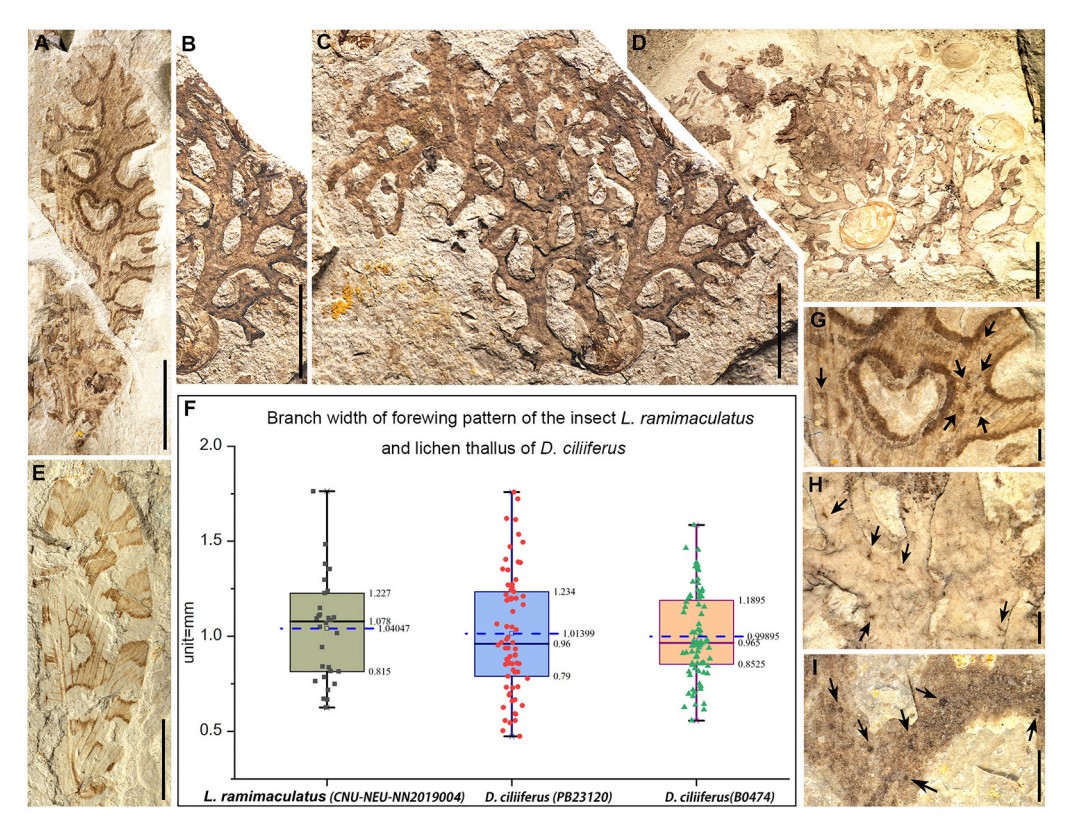

**Figure 4.** The lichen mimicking lacewing *Lichenipolystoechotes ramimaculatus* gen. *et* sp. nov. and *L. angustimaculatus* gen. *et* sp. nov., and fossils of the contemporaneous lichen *Daohugouthallus ciliiferus* Wang, Krings *et* Taylor, 2010. (**A**) Photo of part of *L. ramimaculatus*, with a lichen mimicking forewing pattern, CNU-NEU-NN2019004P. (**B–C**) Photos of the lichen thallus *D. ciliiferus*, PB23120; thallus segment in (**B**); and entire thallus in (**C**). Photos A–C are at the same scale. (**D**) Photo of a nearly intact lichen thallus of *D. ciliiferus*, B0474. (**E**) Photo of *L. angustimaculatus* with a lichen mimicking wing pattern; CNU-NEU-NN2016040P. (**F**) Box scatter plots of measurement data displaying lower and upper extremes, lower and upper quartile, median and average (in the blue dotted line) of branch widths of *L. ramimaculatus*'s forewing pattern (CNU-NEU-NN2019004C) and thallus branch widths of lichen *D. ciliiferus* (PB23120, B0474) separately. (Black, red and green dots represent measurement results of branch pattern widths of lichen-mimicking lacewing and thallus widths of the two lichen specimens, respectively.) (**G**) Part of the wing pattern of *L. ramimaculatus*, with irregular wing spots. (**H, I**) Portion of the thallus of *D. ciliiferus*, with irregular spot-like punctiform pycnidia, B0474 (**H**), B0476P (**I**) The dark arrows indicate the spots on wing of *L. ramimaculatus* and thallus of *D. ciliiferus*. Scale bars: 5 mm in A–E, 1 mm in G–I.

The online version of this article includes the following figure supplement(s) for figure 4:

**Figure supplement 1.** Measuring lines on lichen-mimicking *L. ramimaculatus* and lichen *D. ciliiferus*.

along the wing center and as U-shaped extensions occurring around the wing border. The pigmented branch pattern of the wings has uneven widths and is angulated outwardly. The variation in width of each forewing vein branch conforms well to the variation in width of the lichen's branches, indicating a morphological similarity between the wing markings and lichen thallus (*Figure 4F*; *Figure 4—figure supplement 1*; *Supplementary file 1*: Table S1). Lichens often have punctiform pycnidia (asexual reproductive structures) with black spots appearing on their thallus, especially in extant foliose lichen families such as Parmeliaceae (*Thell et al., 2012*). In *Daohugouthallus ciliiferus* specimens, punctiform black spots occur, but whether they are pycnidia is uncertain. It is noteworthy that a specimen of *L. ramimaculatus* displays similar, scattered spots on its wings that resemble the dark spots on the lichen thallus of *D. ciliiferus* (*Figure 4G–I*), potentially strengthening the similarity between *L. ramimaculatus* and *D. ciliiferus*. Collectively, these details of insect morphology likely enhanced the similarity of the insect with a co-occurring lichen, providing a reasonable inference that the forewing is mimetic with the lichen thallus.

It is generally known that lichens are stable, symbiotic associations of fungi and algae (*Lücking and Nelsen, 2018*). At the same time, lichens are regarded as pioneers in the colonization

of novel surfaces such as bark, rock and soil, which dominate about 7% of the earth's terrestrial surface (*Larson, 1987*), and have a distribution from the polar regions to the tropics (*Lumbsch and Rikkinen, 2017*). They are prominent in arctic-alpine vegetation types in wet and higher montane forests (*Lumbsch and Rikkinen, 2017*). Many extant foliose or fruticose lichens such as taxa of Parmeliaceae are known to be epiphytic or corticolous, and bark surfaces are one of the most common substrates (*Lumbsch and Rikkinen, 2017*). *Daohugouthallus ciliiferus* is considered an epiphytic foliose to subfruticose lichen, and often is found entangled with gymnosperm seed cones (*Figures 1C* and *4D*; *Wang et al., 2010a*). When *Lichenipolystoechotes* moth lacewings reposed in a habitat rich in *D. ciliiferus*, a near perfect match of their appearances would assist their concealment. Among extant Neuroptera, similar appearances of lichen-camouflage or related cases have been recorded in some larvae of green lacewings that carry packets of lichen material on their backs to hide themselves (*Slocum and Lawrey, 1976*; *Wilson and Methven, 1997*). Although *Lichenipolystoechotes* probably lacked the same life-habit as modern lichen-carrying chrysopoid larvae, the Jurassic taxa likely acquired a similar survival advantage when they occupied a lichen-rich habitat. Some extant *Thyridosmylus* species of Osmylidae, another archaic lineage of Neuroptera, possess similar complex wing markings and often occur on moss-laden surfaces of rocks, tree bark and indurated soil surfaces (*Winterton et al., 2017*; *Figure 2B*), which exhibit an impressive consistency with their surroundings (pers. observ. by Yongjie Wang). Although *Lichenipolystoechotes* is a member of Ithonidae, phylogenetically distant to Osmylidae, we infer that their concealment strategy of mimicking cryptogam plants in certain habitats has a deep geochronologic history among ancient lacewing lineages.

Unlike the models of other, co-occurring, plant-mimicking insects, lichen-mimesis of *Lichenipolystoechotes* appears highly specialized (*Figure 5*). Modern lichens can produce a variety of lichenic acids (*Gerson, 1973*) that are unpalatable to many insects and enhance the protective sheltering for animals. Consequently, lichens and lichen-tolerant animals, such as lichen feeding insects and mites, constitute a unique micro-ecosystem. We hypothesize that such a micro-ecosystem existed 165 million-years-ago in Northeastern China that accommodated these trophic, sheltering, defensive and mimetic interactions. Although lichen mimesis is not well documented among extant insects, the most iconic such case of lichen and insect resemblance is the industrial melanism of the peppered moth *Biston betularia* in nineteenth century Britain (*Gerson, 1973*; *Stevens, 2011*). The Industrial Revolution caused elevated levels of soot laden air pollution that resulted in disappearance of lichen shelters for the light-colored morph of *B. betularia*, as their corresponding habitation sites were changed from lightly tinged to dark-hued lichen surfaces that led to their greater vulnerability to predation. This change resulted in the abrupt increase of the dark colored morph of *B. betularia*. When lightly hued lichens returned after aerial pollution was thwarted, *B. betularia* again became dominant as the lightly colored morph. The industrial melanism of *B. betularia* was believed as a textbook example of Darwinian evolution in action, though it was questioned by some authors (*Sargent, 1968*; *Sargent, 1969*; *Coyne, 1998*; *Cook and Saccheri, 2013*). Nevertheless, other studies demonstrated that selection pressures such as predation by birds genuinely affected the differential survival of the pale and dark colored morphs of *B. betularia* under differently hued backgrounds (*Howlett and Majerus, 1987*; *Liebert and Brakefield, 1987*; *Majerus, 2009*; *Walton and Stevens, 2018*). It is possible that the Jurassic *Lichenipolystoechotes* could have gained survival advantage from mimesis of a lichen similar to that of modern *B. betularia*–lichen mimesis. Specifically, if lichen models were present in the habitat occupied by *Lichenipolystoechotes*, survival of the mimic would be assured. It is noteworthy that the winged adults of *Lichenipolystoechotes* would not have been always in the shelter of a lichen model; however, when they were, the conditions of mating, laying of eggs and dispersal would be paramount for survival. If so, high-contrast lichen-like markings could contribute to concealment of the insects. Alternatively, such high-contrast markings of *Lichenipolystoechotes* species also can be interpreted as disruptive coloration, which would confuse the boundaries of moth lacewing and lichen to prevent the detection of a body part essential for survival (*Stevens, 2011*). Consequently, the lichen-like markings of *Lichenipolystoechotes* could likely bring the double protections to the insects-background mimicry and disruptive coloration.

Was there possible benefit to *D. ciliiferus* from its mimetic association with *Lichenipolystoechotes*? This is an open question that could raise multiple alternative explanations. Some modern insects such as ants, dipterans and larva of green lacewings are considered to potentially contribute to dispersal of lichens by transporting lichen propagules to new sites of colonization (*Gerson, 1973*; *Keller and Scheidegger, 2016*; *Ronnås et al., 2017*). In a comparison with such relatively small,

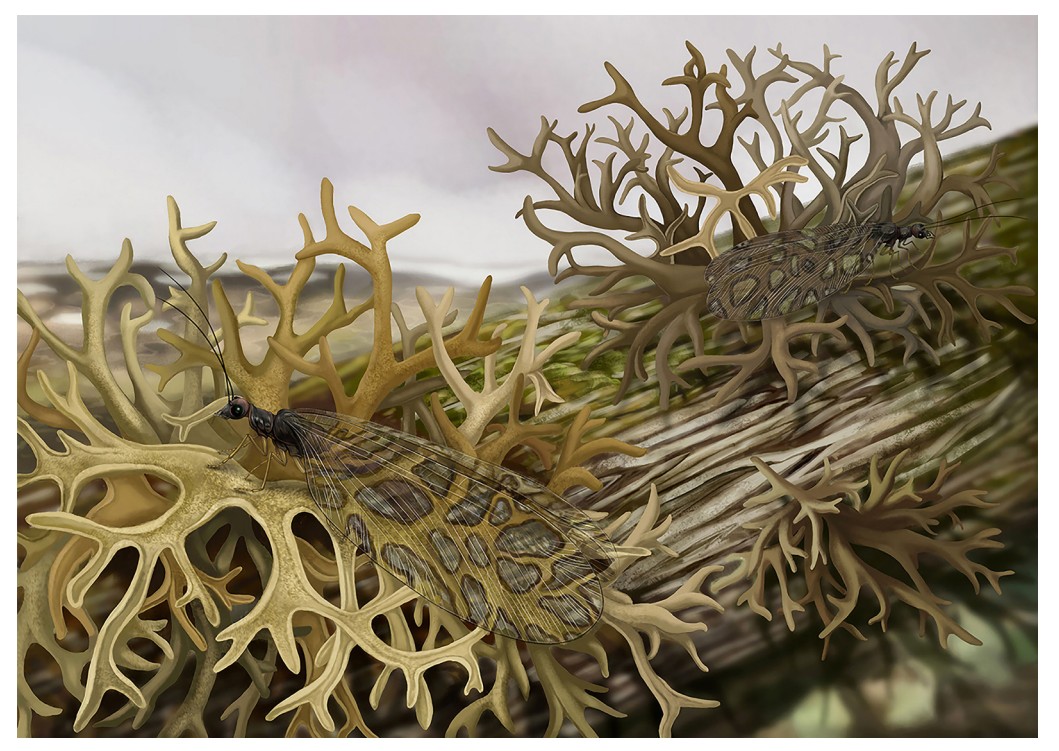

**Figure 5.** Habitus reconstruction of the lichen mimicking lacewing *Lichenipolystoechotes ramimaculatus* gen. *et* sp. nov. on the lichen *Daohugouthallus ciliiferus* Wang, Krings *et* Taylor, 2010. The colors used in the drawing of *D. ciliiferus* is Taupe, referring to the color of extant lichen *Everniastrum cirrhatum*. The body of the *L. ramimaculatus* is reconstructed based on living ithonid species, and the wing is based on the fossil of holotype CNU-NEU-NN2019006P/C. The color of insect is yellowish-brown based on the general coloration of extant polystoechotids. Xiaoran Zuo did the reconstruction drawing.

lichen-carrying insects, *Lichenipolystoechotes* possessed a considerably larger body size that likely was convenient for dispersal of lichen propagules. Notably, sexual reproductive organs such as apothecia have not been found on the *D. ciliiferus* thallus based on light-microscopic morphological and SEM anatomical observations; neither were vegetative propagules such as soredia or isidia seen except along marginal lobules that occasionally were present. This hypothesis of zoochory requires additional evidence for support. However, our alternative hypothesis of benefiting *D. ciliiferus* is based on trophic interactions. As predaceous insects, *Lichenipolystoechotes* inhabited a lichen-rich environment to evade their predators, but they also could have predated and consumed smaller lichen-feeding animals while simultaneously decreasing herbivore damage to the *D. ciliiferus* thallus. This latter hypothesis would require additional verification from evidence of a small ecological web of predator, shelter, defensive and mimetic interactions associated with *Daohugouthallus* and *Lichenipolystoechotes* in the same deposit.

The accepted oldest lichen fossil was reported from the Early Devonian and lichens have existed minimally for 410 million years (*Taylor et al., 1995*; *Honegger et al., 2013*; *Lücking and Nelsen, 2018*), as have the apterygote insects (*Misof et al., 2014*). Both archaic Devonian lineages have evolved more derived, diverse clades of lichens and pterygote insects resulting in a myriad of associations among their modern lineages (*Figure 6*). Although there is virtually no evidence to suggest when and how such association began; in this report, we describe the oldest examples of lichen mimesis that involved two lacewing species resembling a contemporaneous lichen from the same, latest Middle Jurassic deposit. These insect lineages have acquired mimicry association with lichens in less than half of the time (40%) of the duration of both major lineages since the early Devonian (*Figure 6*). This new finding documents a unique survival strategy among mid-Mesozoic Neuroptera, and others await discovery.

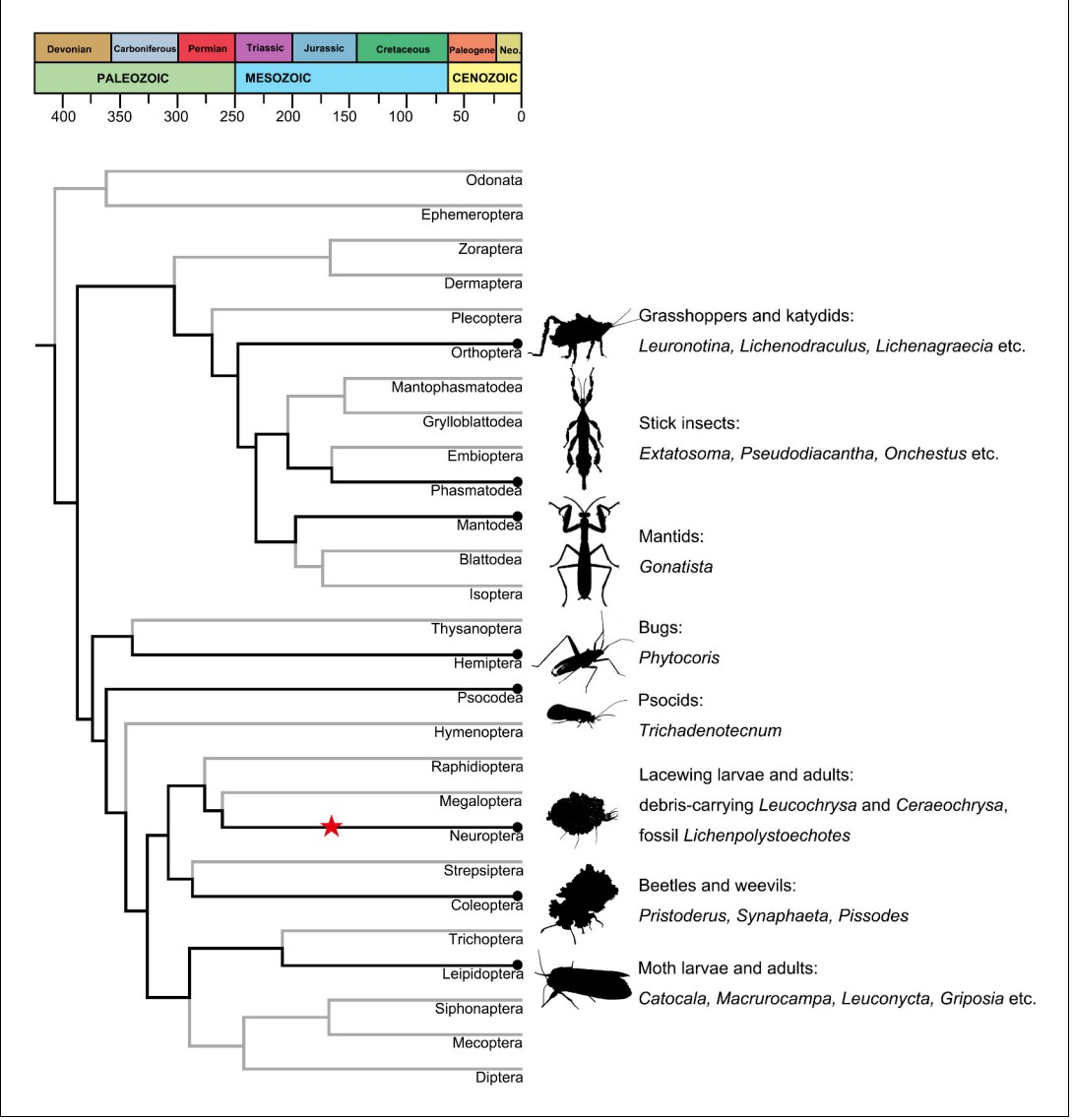

**Figure 6.** Lichen mimicry and camouflage by insects across major insect lineages. Time-dated chronogram based on *Misof et al., 2014*. Specific examples of fossil and modern lichen mimesis by various insect taxa are provided at right. Black dots represent modern insect–lichen-mimetic associations; the star represents the fossil *Lichenipolystoechotes*–lichen mimicry of this study.

# Materials and methods

## Geological context

Specimens were collected from the Daohugou 1 locality of the Jiulongshan Formation, near Daohugou Village, Ningcheng County, approximately 80 km south of Chifeng City, in the Inner Mongolia Autonomous Region, China (119°14.318′E, 41°18.979′N). The age of this formation is 168–152 Ma based on $^{40}$Ar/$^{39}$Ar and $^{206}$Pb/$^{238}$U isotopic analyses (*Hy et al., 2004*; *Liu, 2006*; *Ren, 2019*).

## Specimen repository

CNU-NEU-NN2016040P/C and CNU-NEU-NN2016041 of *Lichenipolystoechotes angustimaculatus* sp. nov., and CNU-NEU-NN2019004P/C of *Lichenipolystoechotes ramimaculatus* sp. nov. are housed in the College of Life Sciences and Academy for Multidisciplinary Studies, Capital Normal University (CNU), Beijing, China. Lichen specimens of *Daohugouthallus ciliiferus* Wang, Krings et Taylor, 2010:

PB23120 is housed in the paleobotanical collection of the Nanjing Institute of Geology and Palaeontology, Chinese Academy of Sciences, in Nanjing, China; B0474 and B0476P/C are housed in the Institute of Vertebrate Paleontology and Paleoanthropology, Chinese Academy of Sciences, in Beijing, China; CNU-LICHEN-NN2019001 and CNU-LICHEN-NN2019002P/C are housed in the Key Lab of Insect Evolution and Environmental Changes, College of Life Sciences and Academy for Multidisciplinary Studies, Capital Normal University, in Beijing, China.

## Experimental methods

The insect and lichen fossils were examined and photographed using a Nikon SMZ25 microscope attached to a Nikon DS-Ri2 digital camera system at the Key Lab of Insect Evolution and Environmental Changes at Capital Normal University in Beijing, China. Lichen compression specimens from the Daohugou one locality were soaked in water for several seconds, dried on filter paper, and then a fragment was lifted up by the edge of a razor blade. One isolated, dried slice was examined and photographed using a Zeiss Axioscope2 compound microscope attached to a Nikon D5100 digital camera system at the State Key Laboratory of Mycology, Institute of Microbiology, at the Chinese Academy of Sciences in Beijing. That piece of lichen fossil then was sputter-coated with gold particles using an Ion Sputter E-1045 (HITACHI), and SEM images were recorded using a scanning electron microscope (Hitachi SU8010) with a secondary electron detector operated at 5.0 kV. Overlay drawings were prepared by Corel DRAW. Box plots were made with Origin 2018 software, which is used to display the distribution of the data of branch width of *L. ramimaculatus*'s forewing pattern and lichen thallus of *D. ciliiferus*. The box plots are formed by two quartiles showing the high frequency of values, and the upper and lower points of the boxes are the maximum and minimum values. All figures were composited in Adobe Photoshop.

## Acknowledgements

We sincerely thank Dr. George Perry (Editor), Dr. Robert Lücking (Reviewer), Dr. Enrique Peñalver (Reviewer), and another anonymous reviewer for their critical comments and constructive suggestions to improve this paper. We are grateful to Dr. Chong Dong (Nanjing Institute of Geology and Palaeontology, Chinese Academy of Sciences) for providing *Figure 4C* and Dr. Boyang Sun (Institute of Vertebrate Paleontology and Paleoanthropology, Chinese Academy of Sciences) for assisting the loan of lichen specimens B0474 and B0476P/C. We thank Xiaoran Zuo for drawing the habitus reconstruction picture in *Figure 5*. We also thank Xuedong Li (College of Life Sciences and Academy for Multidisciplinary Studies, Capital Normal University) for assisting us in the analysis of fossil lichens. This report is contribution 382 of the Evolution of Terrestrial Ecosystems at the National Museum of Natural History in Washington, D.C.

## Additional information

### Funding

| Funder | Grant reference number | Author |
|---|---|---|
| National Natural Science Foundation of China | 31970383 | Yongjie Wang |
| National Natural Science Foundation of China | 31730087 | Dong Ren |
| National Natural Science Foundation of China | 31770022 | Xinli Wei |
| Natural Science Foundation of Beijing Municipality | 5192002 | Yongjie Wang |
| Academy for Multidisciplinary Studies of Capital Normal University | | Dong Ren Yongjie Wang |
| Capacity Building for Sci-Tech Innovation - Fundamental Scientific Research Funds | 19530050144 | Yongjie Wang |

| Program for Changjiang Scholars and Innovative Research Team in University | IRT-17R75 | Dong Ren |
| Support Project of High Level Teachers in Beijing Municipal Universities | IDHT20180518 | Dong Ren |
| Graduate Student Program for International Exchange and Joint Supervision at Capital Normal University | 028175534000 | Hui Fang |
| National Natural Science Foundation of China | 41688103 | Dong Ren |
| Graduate Student Program for International Exchange and Joint Supervision at Capital Normal University | 028185511700 | Hui Fang |

The funders had no role in study design, data collection and interpretation, or the decision to submit the work for publication.

### Author contributions

Hui Fang, Conceptualization, Data curation, Formal analysis, Investigation, Visualization, Methodology, Writing - original draft, Writing - review and editing; Conrad C Labandeira, Conceptualization, Supervision, Validation, Investigation, Writing - review and editing; Yiming Ma, Investigation, Visualization, Writing - original draft; Bingyu Zheng, Investigation, Visualization, Methodology, Writing - original draft; Dong Ren, Conceptualization, Resources, Supervision, Funding acquisition, Validation, Investigation, Project administration; Xinli Wei, Data curation, Formal analysis, Funding acquisition, Investigation, Methodology, Writing - review and editing; Jiaxi Liu, Conceptualization, Resources, Supervision, Validation, Investigation, Project administration; Yongjie Wang, Conceptualization, Resources, Data curation, Formal analysis, Supervision, Funding acquisition, Validation, Visualization, Writing - original draft, Project administration, Writing - review and editing

### Author ORCIDs

Yongjie Wang (iD) https://orcid.org/0000-0003-1397-8481

### Decision letter and Author response

Decision letter https://doi.org/10.7554/eLife.59007.sa1
Author response https://doi.org/10.7554/eLife.59007.sa2

## Additional files

### Supplementary files

• Supplementary file 1. Table S1 Branch width of forewing pattern of *Lichenipolystoechotes ramimaculatus* and lichen thallus of *Daohugouthallus ciliiferus*.

• Transparent reporting form

### Data availability

All data generated or analysed during this study are included in the manuscript and supporting files.

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
