## [Decision Letter]

Thank you for submitting your article "Lichen mimesis in mid-Mesozoic lacewings" for consideration by *eLife*. Your article has been reviewed by George Perry as the Senior Editor, a Reviewing Editor, and three reviewers. The following individuals involved in review of your submission have agreed to reveal their identity: Robert Lücking (Reviewer #1); Enrique Peñalver (Reviewer #2).

The reviewers have discussed the reviews with one another and the Editor has drafted this decision to help you prepare a revised submission.

We would like to draw your attention to changes in our revision policy that we have made in response to COVID-19 (https://elifesciences.org/articles/57162). Specifically, when editors judge that a submitted work as a whole belongs in *eLife* but that some conclusions would require additional new data, as they do with your paper, we are asking that the manuscript be revised to limit claims to those supported by data in hand.

Summary:

This manuscript describes two new fossils of lacewings from the Jurassic, establishing a new genus. The wing pattern in these fossils strikingly mimics another fossil from the same stratum, interpreted by the authors as a lichen. The strengths of the manuscript lie in establishing this potential mimetic association, as well as the underlying analysis and rendering in the form of striking imagery and illustrations. The specimens studied are impressive in their fine preservation. Reviewers praised the work as an important discovery and research, writing that the new vegetal mimesis in Mesozoic insects is fascinating, and that it is a superb contribution to the knowledge of the paleobiology and evolution of the insects.

Essential revisions:

1) Consensus among reviewers was that the lichen affinity of Daohugouthallus ciliiferus was inadequately demonstrated. The authors do present micrographs to support their interpretation, but in my view, these are not convincing, as there is no clear evidence of hyphal structures. See for instance Honegger et al., 2013 for comparison. The authors also interpret the irregular dark dots as pycnidia but provide no sectional evidence, when pycnidia can clearly be identified by their internal anatomy, even in fossils older than 400 my (see Taylor et al., 1994 and Honegger et al., 2013). The evidence for Daohugouthallus being a lichen is at best inconclusive and with at least equal probability it could be a bryophyte or other lower plant. One important argument for Daohugouthallus being a bryophyte rather than a lichen is that these bryophyte morphotypes were around since the Devonian, whereas macrolichens with such a morphology did not evolve until after the K-Pg boundary, i.e. 100 my later than the fossil (see references in the annotations).

1a) For publication in *eLife*, we would require a much more careful assessment of the biological interpretation of Daohugouthallus. It is ok for the authors to state that they believe that this may be a lichen, but may also be something else. This does not detract from the strength of the paper, as interpretation of the mimesis does not depend on the biological interpretation of Daohugouthallus, but the discussion should be adjusted accordingly.

1b) However, this does mean that choice of the generic name Lichenipolystoechotes is over-speculative. Furthermore, reviewers felt that while the new genus can readily be assigned to the polystoechotid genus-group of Ithonidae, the polystoechotid genus-group is not a stable taxonomy taxon, and ID to the family level would be better.

1c) Related to this, at present non-paleobotanists would have difficulty interpreting Figure 4 and the associated discussion; expanded descriptions that potentially could include comparisons with extant correlates would benefit a broader range of readers as the authors developed their (cautious) belief.

1d) Finally, although this approach may be changed completely in revision, we note that it is not clear what the authors mean by "Supplemental Diagnosis". Diagnoses are very conservative elements in taxonomy. In theory, there could be a section named "Original diagnosis" containing exactly the original diagnosis to then comment on it in a subsequent section OR there could be an "Emended diagnosis" containing only the text of a diagnosis (with the correct elements of the original diagnosis mixed with new elements). That "Emended diagnosis" will be diagnosis of this taxon (as all other taxa new research could emended this emended diagnosis, etc.). Of course, after the emended diagnosis a subsequent section can explain the changes done or additions and their relevance. But in the current version of the manuscript, this section is an informal mixture of emended diagnosis and remarks and justification of the changes done.

2) Much of the Discussion section goes into details of extant lichen-related mimicry, mimesis and camouflage and is unrelated to the findings presented in the paper, and otherwise incorrectly interpreted. For instance, lichen mimesis is mostly found in insects that are otherwise not associated with lichens and only frequently co-occur in the same habitat, whereas lichen feeders usually do not exhibit lichen mimesis. There is also no known case where an insect mimicking a lichen would actually aid in its dispersal. Therefore, all this discussion is too speculative or tangential and the discussion should be limited to two aspects: whether Daohugouthallus is in fact a lichen and the evidence leading to the interpretation that the lacewings indeed mimic Daohugouthallus (whether lichen or not), plus giving a context of wing patterns in extant Neuroptera for alternative interpretations. Furthermore, lichens are not plants. Hence the elaboration of the topic of mimicry and mimesis based on plants is misleading. Also, lichens did not evolve into plants as stated at one point.

3) The authors do not offer alternative interpretations for banded wing patterns. These are quite common in diverse insects including extant Neuroptera, and also other animals, but they are not specifically interpreted as lichen mimesis. Rather, such banded patterns represent a general approach to camouflage, by "breaking" the actual shape of the animal. This should be discussed.

4) The relationship between punctiform pycnidia-like dark spots on the talli and minute spots in the wings is not very convincing and the authors could be more conservative in the presentation of this topic. It is difficult to conclude that the small spots in the wings were generated by natural selection during the evolution of this mimetic case when considering the comparative scales in respect to the diaphanous fenestrae with dark limits and these punctiform dark spots. Maybe, the minute spots were visually irrelevant (and also could be artifacts due to fossilization). Do the authors know if there are studies about the relevance of diverse elements of very different size in mimetic cases in extant biota?

5) The cited references require attention, both to ensure the inclusion of recent literature and to confirm that the cited work supports the topic discussed. Overall, a much more careful literature survey is required.

[Editors' note: further revisions were suggested prior to acceptance, as described below.]

Thank you for submitting your revised article "Lichen mimesis in mid-Mesozoic lacewings" for consideration by *eLife*. The reviewers have discussed the reviews with one another and the Reviewing Editor has drafted this decision to help you prepare a revised submission.

Summary:

The reviewers felt that the new SEM data (which impressed us!) helped to make a now-sufficient case for interpretation as lichen mimesis. Most other essential revisions were also addressed.

Essential revisions:

We still believe that the discussion on the broader biological context of mimesis remains too far-reaching. This discussion is based on studies with extant organisms; while the authors present a fine example of extinct mimesis documented by fossils, there are no biological data to support the notion that any of the additional factors (e.g. feeding on lichens, lichen dispersal, impact of environmental changes on mimetic patterns) took place in these fossil taxa. Therefore, while it is fine to briefly mention these factors in a general summary paragraph on lichen mimesis, it is too strong to infer that any of these actually applied to the fossil example.

---

## [Author Response]

Essential revisions:1) Consensus among reviewers was that the lichen affinity of Daohugouthallus ciliiferus was inadequately demonstrated. The authors do present micrographs to support their interpretation, but in my view, these are not convincing, as there is no clear evidence of hyphal structures. See for instance Honegger et al., 2013 for comparison. The authors also interpret the irregular dark dots as pycnidia but provide no sectional evidence, when pycnidia can clearly be identified by their internal anatomy, even in fossils older than 400 my (see Taylor et al., 1994 and Honegger et al., 2013). The evidence for Daohugouthallus being a lichen is at best inconclusive and with at least equal probability it could be a bryophyte or other lower plant. One important argument for Daohugouthallus being a bryophyte rather than a lichen is that these bryophyte morphotypes were around since the Devonian, whereas macrolichens with such a morphology did not evolve until after the K-Pg boundary, i.e. 100 my later than the fossil (see references in the annotations).

We fully understand the concerns of the reviewers, because the lichen affinity of *Daohugouthallus ciliiferus* is at the core of the proposed mimesis relationship. Indeed, the previous figures of *Daohugouthallus ciliiferus* did not adequately show the distinct structures of hyphae and algae. We conducted an additional SEM analysis of specimens of *Daohugouthallus ciliiferus*, and fortunately obtained a series of SEM figures that clearly illustrated the inner structure of *Daohugouthallus ciliiferus*. Our supplementary description, now added in subsection “Emended diagnosis” is:

“upper cortex conglutinate, comprising one cell layer, very thin, c. 1 μm thick (Figure 2A); algal cells globose to near globose, one-celled, mostly 1.5-2.1 μm in diameter, some in framboidal form, anastomosed (Figure 2A, B, F) by or adherent (Figure 2C–H) to the fungal hyphae with a simple wall-to-wall mycobiont-photobiont interface; fungal hyphae filamentous, some shriveled, septate (Figure 2B, C, G, H), 1.2-1.5 μm wide.”

Obviously, the bryophyte or other lower plants could be excluded for the possibility of having such an inner structure. Moreover, no typical bryophytic-grade reproductive structures of antheridia or archegonia were observed in our material.

The new results well match the criteria of the identification of lichen fossil as proposed by Lücking and Nelsen, (2018):

1) Presence of a mycobiont component recognizable as hyphae. The filamentous hyphae *of D. ciliiferus* is distinctly shown in Figure 2B–F;

2) Presence of a photobiont component in agreement with the morphology of undifferentiated, unicellular or filamentous microalgae. The algal cells of *D. ciliiferus* are shown by dark arrow in Figure 2C–F are structurally similar to Honegger et al.’s results, except for having a smaller size; and

3) A spatially correlated arrangement of both components in a way that suggests stable interaction in the form of an exosymbiosis. In Figure 2, especially in Figure 2B,C,E,F,H, it is clearly shown that algal cells are anastomosed or adherent to the fungal hyphae, which are very similar to Honegger et al.’s results (*cf.* Figure 4C,D,E).

Based on the new evidence, we are convinced that *D. ciliiferus* is a lichen. It is a pity that we did not obtain convincing structures of pycnidia, though we had tried repeatedly. We think this does not affect the lichen identity of *D. ciliiferus*, considering the solid evidence of hyphae and algae, and the frequent rarity of pycnidia in modern lichens. We deleted any corresponding statements of pycnidia in the revised text.

We admit that the discovery of a Jurassic macrolichen will be greatly challenge to the current knowledge on systematics and evolution of lichen. However, the fossil of *D. ciliiferus* are real and clearly corroborates the existence of a Jurassic macrolichen. We note that the fossil record of macrolichens is sparse and that 90 percent of fossil occurrences originate from Paleogene ambers (Lücking and Nelson, 2018). As for the significance of the oldest macrolichen and the evolution of lichen, it is beyond the scope of this paper, but it has been considered as our next project.

1a) For publication in eLife, we would require a much more careful assessment of the biological interpretation of Daohugouthallus. It is ok for the authors to state that they believe that this may be a lichen, but may also be something else. This does not detract from the strength of the paper, as interpretation of the mimesis does not depend on the biological interpretation of Daohugouthallus, but the discussion should be adjusted accordingly.

See the details see the above-mentioned comment. Indeed, there were some improper statements in the original paper, and we have made relevant modifications in the revised text.

1b) However, this does mean that choice of the generic name Lichenipolystoechotes is over-speculative. Furthermore, reviewers felt that while the new genus can readily be assigned to the polystoechotid genus-group of Ithonidae, the polystoechotid genus-group is not a stable taxonomy taxon, and ID to the family level would be better.

*Lichenipolystoechotes* is a typical polystoechotid insect within Ithonidae, based on the particular characters of its venation. Recently, ‘Polystoechotidae’ was synonymized with Ithonidae, as discussed by Winterton and Makarkin, 2010. But this synonymization did not resolve relationships within Ithonidae, and whether Polystoechotinae is a subfamily of Ithonidae, or of some lower rank. As the polystoechotid genus-group was tentatively used for previous fossil ‘polystoechotid lacewings’, our assignment of *Lichenipolystoechotes* is nomenclaturally valid at the family level as a member of Ithonidae.

1c) Related to this, at present non-paleobotanists would have difficulty interpreting Figure 4 and the associated discussion; expanded descriptions that potentially could include comparisons with extant correlates would benefit a broader range of readers as the authors developed their (cautious) belief.

Figure 4 was intended to present the comparisons of wing patterns of *Lichenipolystoechotes* species and the thalli of *Daohugouthallus* specimens. In addition to the overall details in the likenesses of the wing pattern and lichen thallus, a quantitative method was used to measure the variation of branch widths of forewing pattern and lichen thalli to further analytically assess their similarity. In the results, the variation of branch widths of *L. ramimaculatus*’s forewing pattern is well in accord with the variation of lichen thalli (Figure 4F, Figure 4—figure supplement 1; Supplementary file 1). We provided the supplement explanations of Figure 4 in both the Discussion section and Materials and methods section of the main text.

1d) Finally, although this approach may be changed completely in revision, we note that it is not clear what the authors mean by "Supplemental Diagnosis". Diagnoses are very conservative elements in taxonomy. In theory, there could be a section named "Original diagnosis" containing exactly the original diagnosis to then comment on it in a subsequent section OR there could be an "Emended diagnosis" containing only the text of a diagnosis (with the correct elements of the original diagnosis mixed with new elements). That "Emended diagnosis" will be diagnosis of this taxon (as all other taxa new research could emended this emended diagnosis, etc.). Of course, after the emended diagnosis a subsequent section can explain the changes done or additions and their relevance. But in the current version of the manuscript, this section is an informal mixture of emended diagnosis and remarks and justification of the changes done.

We sincerely appreciate the rigorous review from the reviewer and Editor that address this question. In the text, we have added new evidence of the presence of hyphae and algae within the *Daohugouthallusciliiferus* structure, which corroborates its lichen affinity. However, in our view, a ‘Supplemental Diagnosis’ should be used when there are major errors in the original diagnosis. Rather, we have combined the mixed contents of elements of the original diagnosis with our supplementary elements into subsection “Emended diagnosis” and have used the subsection “Remarks” to explain these new findings.

2) Much of the Discussion section goes into details of extant lichen-related mimicry, mimesis and camouflage and is unrelated to the findings presented in the paper, and otherwise incorrectly interpreted. For instance, lichen mimesis is mostly found in insects that are otherwise not associated with lichens and only frequently co-occur in the same habitat, whereas lichen feeders usually do not exhibit lichen mimesis. There is also no known case where an insect mimicking a lichen would actually aid in its dispersal. Therefore, all this discussion is too speculative or tangential and the discussion should be limited to two aspects: whether Daohugouthallus is in fact a lichen and the evidence leading to the interpretation that the lacewings indeed mimic Daohugouthallus (whether lichen or not), plus giving a context of wing patterns in extant Neuroptera for alternative interpretations. Furthermore, lichens are not plants. Hence the elaboration of the topic of mimicry and mimesis based on plants is misleading. Also, lichens did not evolve into plants as stated at one point.

First, we should express our thanks for the reviewers’ critical comments that provided some new ideas to improve this paper. The mimicry cases of fossil insects are scarce, and most cases belong to vascular plant mimicry (absence of lichen mimicry). Thus, when we introduce an overview of relevant modern mimicry, plant mimicry was necessarily mentioned. We agree with the reviewer’s criticism that the ‘plant’ was incorrectly used in some places, and we had corrected this in the revised text.

As for the taxonomic attribution of *Daohugouthallus*, it was discussed in Issue 1 above.

As for the lichen mimesis of insects, we consider it should represent a structural specialization of insects to adapt to the lichen-covered backgrounds. When one insect possessing lichen-like pattern rests on an appropriate, lichen-covered background, we maintain that insects could obtain the survival advantages from this relationship, as *Lichenipolystoechotes* undoubtedly did. Consequently, such an association can validly be treated as a lichen mimesis. Of course, not all insects interacting with lichens evolve into lichen mimics, and correspondingly they can evolve other biological or behavioral adaptations for survival. We have provided hopefully some clarity in explaining this in the revised paper.

With regard to assistance in the dispersal of the lichen by *Lichenipolystoechotes*, we have noted extant cases, sourced in the literature, where insects such as lichen-feeding ants and lichen-camouflaged larva of lacewings, could contribute to the dispersal of lichens through the transportation of propagules. As Jurassic *Lichenipolystoechotes* were winged insects that frequented lichens, we infer that considerable potential was present to contribute to lichen dispersal, akin to extant insects. We agree with the reviewer’s criticism that there is no direct evidence to support this, and we have adjusted this section in the revised paper to reflect this concern. Nevertheless, we aver that our inference of the possibility of such an association is valid.

3) The authors do not offer alternative interpretations for banded wing patterns. These are quite common in diverse insects including extant Neuroptera, and also other animals, but they are not specifically interpreted as lichen mimesis. Rather, such banded patterns represent a general approach to camouflage, by "breaking" the actual shape of the animal. This should be discussed.

This is an important suggestion. The wing markings of *Lichenipolystoechotes* have a very distinctive condition of (i) banding paralleling longitudinal veins, (ii) particularly shaped diaphanous fenestrae, (iii) a unique distribution of colors and hues, and other features that are quite different than ordinary, run-of-the-mill wing banding in fossil and extant Neuroptera. Nevertheless, we have added alternative explanations in the revised text.

4) The relationship between punctiform pycnidia-like dark spots on the talli and minute spots in the wings is not very convincing and the authors could be more conservative in the presentation of this topic. It is difficult to conclude that the small spots in the wings were generated by natural selection during the evolution of this mimetic case when considering the comparative scales in respect to the diaphanous fenestrae with dark limits and these punctiform dark spots. Maybe, the minute spots were visually irrelevant (and also could be artifacts due to fossilization). Do the authors know if there are studies about the relevance of diverse elements of very different size in mimetic cases in extant biota?

We initially suspected these spots likely to be pycnidia. However, the evidence to reflect that these spots were pycnidia could not be obtained. We agreed with the reviewer’s comments and have accordingly adjusted the revised text.

5) The cited references require attention, both to ensure the inclusion of recent literature and to confirm that the cited work supports the topic discussed. Overall, a much more careful literature survey is required.

We accepted the criticism of the reviewers, and the references have been checked out and updated.

[Editors' note: further revisions were suggested prior to acceptance, as described below.]

Essential revisions:We still believe that the discussion on the broader biological context of mimesis remains too far-reaching. This discussion is based on studies with extant organisms; while the authors present a fine example of extinct mimesis documented by fossils, there are no biological data to support the notion that any of the additional factors (e.g. feeding on lichens, lichen dispersal, impact of environmental changes on mimetic patterns) took place in these fossil taxa. Therefore, while it is fine to briefly mention these factors in a general summary paragraph on lichen mimesis, it is too strong to infer that any of these actually applied to the fossil example.

We understand the reviewer’s concern. Among studies of fossils, methodologically it is common to reconstruct or infer the biologies, behaviors, and ecological associations of the past organisms based on what we know about living descendant lineages, as well as the available evidence from the fossils themselves. This is the subject matter of paleoecology; and it includes various lines of evidence, such as functional morphology, trace fossils associated with the behaviors of their makers, evidence of various types of interactions among fossils (herbivory, pollination and mimicry come to mind), but also includes ecological uniformitarianism—that the ecologies of modern lineages are relevant for understanding the biologies of their ancestors in those same lineages. To illustrate this latter point, how would one interpret the prolonged saber-like teeth of an extinct lineage of cat-like carnivores, such as Smilodon, without assessing similar developments of such teeth (albeit less extreme) in various modern carnivores? This is a common principle in paleoecology; otherwise, the interpretation of the fossil record becomes a useless exercise since the species we are attempting to understand are not extant. Our reconstruction of the ecology of the fossil should be considered as a hypothesis to be tested from additional evidence. Our inference of “additional factors” is based on deductive reasoning that is informed by ideas, assumptions, general principals and facts, and is presented as a hypothesis to be tested with additional fossil and modern material. We judge that it is necessary to present additional ideas, albeit some based on inference, to our audience in understanding our finding in a broader ecological context. Consequently, we have conditioned our interpretations by presenting them as a hypothesis for further testing. The major modifications were listed as follows:

1) Because we did not obtain definitive evidence for lichen propagules and lichen damage, we modified the corresponding statements about feeding on lichens and lichen dispersal in the revised manuscript. As mentioned in the previous revised version of the manuscript, we considered that the existence of Jurassic lichens probably was supported by the presence of a confined ecological web that would require documentation of lichens and their mimics, and other trophic or trophic-related interactions, as is the case among extant lichen–mimicry–insect webs (another example of ecological uniformitarianism). However, documentation of such a micro-ecological web would require additional evidence.

There is another point to be made in the above context. We note that in paleontology, the oldest known occurrence of a fossil *almost always* is not the earliest actual occurrence of that fossil. This is due to the vagaries of the fossil record, lack or interest or the number of practitioners of the particular fossil group in question, or mistakes in taxonomic identifications. It is almost certain that there are lichens that are older than the Jurassic. We would like to assert that the previous absence of Jurassic lichens is not a valid reason for establishing the existence of Jurassic lichens, should the fossil record provide positive evidence. We will continue to explore for the presence of Jurassic lichens, and it is hoped that more evidence will be found.

2) As for the “impact of environmental changes on mimetic patterns” passage, this statement had been deleted in the revised manuscript. Because the function of lichen mimesis in *Lichenipolystoechotes* is virtually same as that of *Biston betularia*, we cited the classic case to stress the importance of lichen mimesis in a modern context. Accordingly, it would have been easier for an audience to understand our findings.